# Vasovagal Reactions during Interventional Pain Management Procedures—A Review of Pathophysiology, Incidence, Risk Factors, Prevention, and Management

**DOI:** 10.3390/medsci10030039

**Published:** 2022-07-25

**Authors:** Brian Malave, Bruce Vrooman

**Affiliations:** 1Geisel School of Medicine at Dartmouth, Hanover, NH 03756, USA; 2Section of Pain Medicine, Department of Anesthesiology, Dartmouth-Hitchcock Medical Center, 1 Medical Center Drive, Lebanon, NH 03756, USA; bruce.m.vrooman@hitchcock.org

**Keywords:** vasovagal reaction, epidural spinal injection, interventional pain management procedure, antimuscarinic, moderate sedation, anxiolytic

## Abstract

Vasovagal reactions are a benign but common outcome of interventional pain management procedures that can negatively impact patient care, including aborted procedures and fear of future procedures that would otherwise help the patient. Research has been done on the incidence, risk factors, and management of vasovagal reactions resulting from such procedures, but less is known about how to prevent these reactions from occurring. In this paper, we present a literature review of the pathophysiology, incidence, risk factors, prevention, and management of vasovagal reactions during interventional pain management procedures, with an emphasis on the relative lack of research and conflicting advice on preventive measures. We found that moderate sedation and anxiolytics have been used prophylactically to prevent vasovagal reactions, but their side-effect profiles prevent them from being used commonly. Less studied is the prophylactic administration of antimuscarinics and IV fluids, despite the potential benefit of these measures and relatively low side-effect profile. We explore these topics here and offer advice for future research to fill the gaps in our knowledge.

## 1. Introduction

Vasovagal reactions—defined as a rapid drop in heart rate and/or blood pressure, usually in response to a stressful trigger—are a common complication of interventional pain management procedures. Three types of vasovagal responses have been described in the literature: a cardioinhibitory form (HR < 40 bpm), vasodepressor form (SBP < 80 mmHg or decrease by >30% without significant HR reduction), or mixed form (HR < 40 bpm and SBP < 80 mmHg or decrease by >30%) [1]. Typical symptoms of a vasovagal reaction are lightheadedness or dizziness, palpitations, weakness, blurred vision, nausea, feelings of warmth or coldness, and sweating. When a vasovagal reaction results in a loss of consciousness, it is termed vasovagal syncope. Although vasovagal reactions are usually benign in nature, they can lead to more serious complications for both patients and providers, such as aborted procedures, cardiac arrythmias, or fear of future procedures [2]. It is thus useful for pain medicine clinicians to identify the risk factors, prevention, and management of vasovagal reactions in an outpatient setting.

### 1.1. Aims and Purpose

The primary purpose of this review is to examine what the literature has shown regarding the incidence, pathophysiology, prevention, and management of vasovagal syncope resulting from interventional pain management procedures, with an emphasis on the conflicting advice for preventive measures. Our criteria for interventional pain management procedures include epidural steroid injections, medial bundle branch blocks, radiofrequency ablation, and any other fluoroscopically or ultrasound-guided procedures commonly performed in an outpatient pain management clinic. The secondary purpose of this paper is to examine how vasovagal syncope has been prevented in other clinical settings besides outpatient pain management to explore how these preventive measures may be applicable, and whether they merit further investigation.

### 1.2. Methods

We searched the Google Scholar database with the terms “vasovagal epidural steroid injection”, “vasovagal spine procedure”, “vasovagal prevention”, “vasovagal treatment”, “antimuscarinics vasovagal”, “anxiolytic vasovagal”, and “sedation vasovagal”. We also performed a PubMed search on glycopyrrolate as a preventive measure—since we had a suspicion it might be of use in preventing vasovagal syncope—with the terms “glycopyrrolate (MeSH) OR glycopyrrolate[tiab] OR glycopyrronium[tiab] OR cuvposa[tiab] OR robinu[tiab]”, “syncope, vasovagal (MeSH) OR syncope[tiab] OR vasovagal[tiab] OR vagally mediated[tiab] OR faint*[tiab] OR dizzy[tiab] OR dizzi*[tiab] OR lightheaded[tiab] OR hypotensive event*[tiab] OR bradycardi*[tiab]”, and “injections, spinal (MeSH) OR spine[tiab] OR spinal[tiab] OR injection[tiab] OR epidural[tiab]” (Table 1).

Results were chosen if they were peer-reviewed scientific publications. We chose to write a narrative review rather than a systematic review, since the quality of evidence we found was mixed, and in some cases lacking, making a systematic review more challenging, as suggested by Toljan and Vrooman 2018 [3].

## 2. Pathophysiology

As mentioned previously, “vasovagal” reaction is a term that describes either “vaso-” depression (evident as hypotension), “vagally” mediated cardioinhibition (evident as bradycardia), or a combination of both. When loss of consciousness (i.e., syncope) occurs after a vasovagal reaction, it is term vasovagal syncope. Vasovagal syncope is the most common type of syncope, comprising up to 40% of all outpatient syncopal events. The most common explanation for why vasovagal responses occur is the Bezold–Jarisch reflex. The theory is that excessive venous pooling leads to decreased blood pressure sensed by baroreceptors on the aortic arch, carotid sinus, heart walls, and intrathoracic vessels, which relay the information to the nucleus tractus solitarius of the brain, ultimately inhibiting sympathetic response and increasing vagal tone. This leads to hypotension and bradycardia. This theory is supported by tilt-table testing, in which a patient lies down on a table and is then tilted upright at a specified angle, leading blood to pool in the leg veins secondary to gravity. If the patient exhibits a vasovagal response, it is likely due to the Bezold–Jarisch reflex [4].

The pathophysiology of vasovagal reaction to interventional pain management procedures specifically is multifactorial and not entirely understood. For epidural steroid injections, some have proposed that the epidural anesthesia may lead to sympathetic blockade, resulting in lower venous tone and decreased cardiac output [5,6]. Other known triggers of vasovagal reaction that may occur during interventional pain management procedures—but not specific to these procedures—are fear, anxiety, disgust, pain, imagined or real exposure to bodily harm, sight of blood, hunger, heat, and others. In all cases, these external stimuli lead to increased cardiac contractility despite an underfilled left ventricle, which is sensed by mechanoreceptors in the ventricle and relayed to the brain via vagal afferent nerves. The brain responds to the stimuli by increasing parasympathetic tone, leading to bradycardia and hypotension, much like in the Bezold–Jarisch reflex [7].

## 3. Incidence

Vasovagal reaction is the most common immediate adverse event of interventional pain management procedures, with reported rates ranging from 0 to 4% in the literature [2,6,8,9,10,11,12,13]. Kennedy et al., 2013 [8] reported an overall vasovagal reaction rate of 2.6% for over 8000 fluoroscopically guided interventional procedures performed over 5 years, but a higher vasovagal rate of 3.5% for epidural steroid injections specifically. There does not appear to be a difference in vasovagal rate for transforaminal vs. interlaminar injections.

Differences have been reported for cervical vs. lumbar epidural steroid injections. Trentman et al., 2009 [11] reported that cervical epidural spinal injections (CESI) were seven times as likely as lumbar epidural spinal injections (LESI) to cause vasovagal reactions (8% vs. 1%, respectively), a finding aligned with their anecdotal observations. The study examined all epidural injections performed by eight fully trained staff physicians from 1996 to 2005, and matched each LESI to a CESI performed by the same physician to try to maintain uniformity. The authors suggested that having a flexed neck in a prone position, the use of head drapes, anxiety about having a procedure done near the neck, and direct stimulus from the cervical procedure all contributed to higher vasovagal reactions than lumbar injections.

Interestingly, Kennedy et al., 2013 [8] reported a lower CESI vasovagal rate of 0.47% compared to an LESI vasovagal rate of 3.67%, contradicting the results of Trentman et al., 2009 [11]. The authors hypothesized that CESIs are more superficial than LESIs and thus less likely to elicit a vasovagal response. However, they admitted that there were various confounding variables in their study, including better-trained fellows performing CESIs only after they had mastered performing LESIs.

## 4. Risk Factors

Several risk factors have been linked to the development of vasovagal reactions secondary to interventional pain management procedures. The most important risk factor is a history of prior vasovagal reaction. Patients who have had vasovagal reactions in the past are more likely to experience future vasovagal reactions [13,14]. Other risk factors of vasovagal reactions are male sex, age < 65 years, and preprocedure pain score. Kennedy et al., 2013 [8] found that males were twice as likely as females to have vasovagal reactions to interventional pain management procedures. The reasons are unclear, but the authors proposed that greater muscle mass may produce more precipitous decrease in blood pressure. Both males and females <65 years old had 2.4 times the odds of vasovagal reaction of patients >65 years old. The authors suggested that older patients may have more experience with previous medical interventions and have a higher threshold for pain. Lastly, Kennedy et al., 2013 [8] found that patients with a preprocedure pain score of <5 on a 10-point scale had a vasovagal reaction rate of 3.2% compared to 2.2% for patients with a pain score of >5. They proposed that having higher baseline pain may translate into having a higher threshold for pain and discomfort during interventional pain management procedures. Said another way, patients with lower pain at baseline might have a lower threshold for discomfort during interventional pain management procedures and therefore a lower threshold for developing a vasovagal reaction during the procedure. Other risk factors of vasovagal reaction proposed in the literature are baseline hypotension, bradycardia, dehydration, or anxiety [13,15].

Aside from patient demographics, another risk factor that impacts likelihood of vasovagal reactions is degree of training of the physician performing the procedure. Schneider et al., 2014 [16] retrospectively analyzed 4482 transforaminal epidural spinal injections from March 2004 to January 2009 and found a vasovagal reaction rate of 2.7% when the procedure was performed by an attending physician, 4.1% when performed by a fellow, and 5.5% when performed by a resident, suggesting that higher levels of training lead to less likelihood of vasovagal reactions in the patient. As mentioned previously, Kennedy et al., 2013 [8] reported a lower CESI vasovagal rate (0.47%) than the LESI rate (3.67%), acknowledging that providers performing CESIs had already mastered performing LESIs (Table 2).

## 5. Prevention

### 5.1. Sedation

The American Society of Anesthesiologists defines the continuum of sedation as minimal, moderate, deep, or general anesthesia. Minimal sedation allows the patient to be responsive to verbal stimuli, and is usually achieved with oral medications or nitrous oxide (“laughing gas”). In moderate or “conscious” sedation, the patient feels drowsy and may fall asleep, but awakens to verbal/tactile stimulation. This is usually achieved with IV medications. Deep sedation occurs when the patient is asleep through the procedure, but awakens to painful stimulation. General anesthesia occurs when the patient is unwakeable, even with painful stimuli [17].

Some evidence suggests that moderate sedation might reduce risk of vasovagal reaction, especially in patients with a history of vasovagal reactions. Kennedy et al., 2015 [2] performed 6364 spine injections, 214 of which were done with conscious sedation with midazolam and fentanyl. They found that none of the injections performed with sedation led to vasovagal reaction, while 3.3% of injections done without sedation led to vasovagal reaction. They analyzed the data further and found that 134 injections had been done on patients with a history of vasovagal reactions. Of these, 90 were done without sedation and 44 with moderate sedation. Those who received sedation did not experience any recurrent vasovagal reaction (vasovagal rate of 0), while those who did not receive sedation had a recurrent vasovagal reaction rate of 23.3%, suggesting that conscious sedation may be an effective measure to prevent the recurrence of vasovagal reactions.

However, sedation during interventional pain management procedures is associated with several risks. A sedated patient may not necessarily experience pain if a spinal nerve or the spinal cord is inadvertently contacted, and thus may not necessarily provide reliable feedback to the proceduralist who is attempting to ensure that there is no central neurological damage after the lidocaine test injection [9,18,19]. There are reported cases of traumatic spinal cord injury during interventional spine procedures in which sedation was used [20]. The frequency at which these iatrogenic injuries occur—and the role of sedation in predisposing to these injuries—is debated. Schaufele et al., 2011 [21] examined 2494 interventional spine procedures—half performed under conscious sedation and the other half without sedation—and found no significant difference in rates of adverse events at 1 day and 3 days postoperation. However, Rathmell et al., 2011 [22] examined ASA malpractice closed claims from 2005 to 2008 and found that 67% of cervical procedure claims associated with spinal cord injury involved the use of sedation or anesthesia. Of these claims, 25% of patients were nonresponsive during the procedure, indicating that they could not provide reliable feedback to the provider. Other complications of sedation include airway compromise, arrhythmia, hypotension, venous thrombosis, pulmonary embolism, nausea, vomiting, allergic reactions, and even death [9]. The possibility of putting patients’ health at risk makes sedation a less favorable choice for prevention of vasovagal syncope.

In summary, sedation has been shown to be an effective measure to prevent the recurrence of a vasovagal reaction in patients with a history of such. However, the routine use of sedation as a primary preventive measure for vasovagal reactions is likely not recommended, given its known risks and the overall low likelihood of a vasovagal reaction (Table 3).

### 5.2. Anxiolytics

Anxiety is a well-established risk factor of vasovagal reactions, and fear of procedures increases the likelihood of a vasovagal response [15,23]. As such, anxiolytic medications, such as benzodiazepines, have been used in a variety of clinical settings—ranging from outpatient breast biopsy to dermatological procedures—to reduce preoperative anxiety levels [24,25,26]. Benzodiazepines have been shown not only to lower anxiety before procedures but also to lower vasovagal reaction rates during the procedure. Gebhardt et al., 2018 [27] retrospectively examined the charts of 2747 patients undergoing low-dose intrathecal anesthesia during outpatient procedures, with 1291 patients receiving anxiolytic premedication of 1–2 mg IV midazolam. Vasovagal reaction rates were 15% for patients who did not receive midazolam and 7.5% for those who did (*p* < 0.001), suggesting that benzodiazepines lower vasovagal rates [27]. James et al., 2005 [28] similarly found that giving 2–4 mg sublingual lorazepam to women undergoing stereotactic breast biopsies successfully prevented a recurrent vasovagal reaction in 95% who had previously had a vasovagal reaction during prior biopsy. The use of benzodiazepines for vasovagal reactions resulting specifically from interventional pain management procedures has not been studied to our knowledge, but merits research given its apparent benefit for lowering vasovagal reactions in other clinical settings. 

There are side effects of benzodiazepines that might make them less favorable to use during interventional pain management procedures. Benzodiazepines are known to cause sedation and increase the risk of motor vehicle accidents, especially in the elderly [29,30], which might prolong the time to achieve readiness for discharge. However, patients undergoing interventional pain management procedures under local anesthesia are not advised to drive for at least 12 h after their injection, even if sedation or anxiolytics are not used [31]. Gebhardt et al., 2018 [27] found that administration of 1–2 mg midazolam IV prior to low-dose intrathecal anesthesia for various outpatient procedures did not prolong time to achieve readiness for discharge. More studies should be done on the effects of benzodiazepines specifically for interventional pain management procedures, but we suspect the time to discharge should not be affected.

Other side effects of benzodiazepines include confusion, anterograde amnesia, agitation, and increased risk of falling, all of which are increased in elderly persons [32,33]. Benzodiazepines have also been linked to teratogenic effects and poor outcomes on fetal health [33,34], although interventional pain management procedures are generally avoided in pregnancy unless conservative measures fail [35,36]. These side effects should be taken into consideration when determining whether to administer benzodiazepines to a patient prior to an interventional pain management procedure (Table 4).

### 5.3. Antimuscarinics

Another pharmacological agent that has the potential to prevent vasovagal reactions is an antimuscarinic. To understand this, one must understand the pathophysiology of the vasovagal response. The vasovagal response is a reflex arc within the parasympathetic nervous system (PNS) that uses acetylcholine as its main postganglionic neurotransmitter. Thus, pharmacologic agents that block the effects of acetylcholine at its “muscarinic” receptor—i.e., antimuscarinics—should be expected to both treat and prevent the vasovagal response. 

Atropine is an alkaloid extract and antimuscarinic agent derived from nightshade plants, including *Atropa belladona* (also known as “deadly nightshade”), Jimson weed, and mandrake. It can be administered ophthalmologically as a mydriatic agent or more commonly intravenously or intramuscularly for treatment of cholinergic crisis, symptomatic bradycardia, and inhibition of salivation and secretions during surgical procedures. Side effects include tachycardia, dry mucous membranes, anhidrosis, urinary retention, and constipation. Glycopyrrolate, a quaternary ammonium drug, is another antimuscarinic agent similar to atropine that is used primarily to inhibit salivary, tracheobronchial, and pharyngeal secretions preoperatively during induction of anesthesia and intubation [37,38,39]. Comparatively, it has greater potency and longer duration of action than atropine. An older study demonstrated differences in end-organ effects between atropine and glycopyrrolate. Glycopyrrolate had a selective and prolonged inhibitory effect at salivary and sweat glands, with minimal cardiovascular, ocular, and CNS effects compared to atropine [40].

Antimuscarinic agents have been studied in the treatment of vasovagal reactions. Santini et al., 1999 [41] demonstrated the efficacy of atropine in treating vasovagal symptoms in patients with the cardioinhibitory form of vasovagal syncope (i.e., characterized by bradycardia <40 bpm without significant blood pressure drop, as defined in our introduction). A selection of patients underwent the tilt test, in which they lay down on a table and were then tilted upright at a specified angle, leading blood to pool in the leg veins and potentially trigger a vasovagal response via the Bezold–Jarisch reflex. Patients with a positive tilt test underwent a second tilt test within 2 weeks of the first diagnostic test, and those with a negative second test were excluded from the group. Eighty-four patients with two positive tilt tests were divided into two groups—placebo or atropine at 0.02 mg/kg. After a repeat tilt-table test, symptoms resolved in 69.7% of patients administered atropine compared to 21.9% of patients on placebo, provided that their heart rate was less than 40 bpm, demonstrating the efficacy of atropine in resolution of cardioinhibitory vasovagal syncope. Atropine did not resolve vasovagal symptoms in patients with vasodepressor syncope, however (defined as significant drop in BP without bradycardia, as described in our introduction), suggesting that it may have more cardiac than vasopressor effects. 

Antimuscarinic agents have not only been studied for the treatment but also the prevention of vasovagal reactions. Prophylactic administration of atropine and glycopyrrolate has been demonstrated to lower vasovagal reaction rates in several procedures, including removal of femoral arterial sheaths [42], cryoballoon ablation in patients with atrial fibrillation [1], C-sections [43], and ophthalmological surgeries [44,45]. Most of these studies have not described which type of vasovagal reaction is prevented by antimuscarinics—i.e., cardioinhibitory, vasodepressor, or mixed. Of those that did, Sun et al., 2017 [1] found that preoperative administration of atropine prior to cryoballoon ablation for atrial fibrillation prevented all three forms of vasovagal syncope. However, Chamchad et al., 2011 [43] found that preoperative glycopyrrolate was effective in the prevention of bradycardia, with minimal to no effect on blood pressure, for women undergoing C-sections. More research needs to be done to clarify these conflicting results. It would also be useful to determine if antimuscarinics given in conjunction with IV fluids are more effective in resolving vasodepressor or mixed forms of syncope than antimuscarinics alone, since IV fluids should correct volume status.

It is interesting to note that a randomized, placebo-control study on prevention of vasovagal syncope during ophthalmological (squint) surgery found fewer side effects associated with glycopyrrolate than atropine. Mirakhur et al., 1982 [44] randomized 160 children (1–14 years old) undergoing ophthalmological (squint) surgery to receive atropine, glycopyrrolate, or placebo at various doses and routes of administration (IV or IM). They found that IV administration of either glycopyrrolate or atropine significantly lowered rates of oculocardiac reflex (a type of vasovagal syncope defined as reduction in HR by >20%), but that glycopyrrolate was associated with a smaller magnitude of tachycardia than atropine. Yang et al., 1996 [45] similarly wrote that glycopyrrolate is less likely to cause tachycardia or dry mouth than atropine when given for ophthalmological surgeries. The lower side-effect profile of glycopyrrolate may make it a more favorable option than atropine for prevention of vasovagal syncope.

While all the aforementioned studies examined the utility of antimuscarinics in prevention of vasovagal reaction resulting from various procedures, no study to our knowledge has evaluated the utility of prophylactic antimuscarinics specifically for interventional pain management procedures. Mahajan 2008 [46] recommends giving IV glycopyrrolate in increments of 0.2 mg for prevention of vasovagal syncope during interventional pain management procedures for patients with a history of vasovagal episodes, but did not provide references to support his recommendation. This represents a large gap in the literature that merits more attention (Table 5).

### 5.4. Hydration and IV Fluids

The possibility of using sedation, anxiolysis, or antimuscarinic agents to prevent vasovagal reactions raises the question of whether an IV line with fluids running should be placed prior to interventional pain management procedures. This would allow IV medications to be given promptly during the procedure if needed, without needing to stop the procedure to insert an IV line. Additionally, IV fluids in and of themselves might prevent vasovagal syncope for patients with baseline hypotension, bradycardia, dehydration, or anxiety. This is especially important to consider if a patient has fasted prior to the procedure. As per ASA guidelines, patients undergoing procedures with moderate sedation should not have clear liquids or solid foods 2 h or 6 h before their procedure, respectively, to mitigate the risk of airway compromise or aspiration [47]. Patients choosing to receive sedation for prevention of vasovagal reaction may thus be dehydrated at baseline, and IV fluid administration may help to further prevent vasovagal reactions. 

For these reasons, some have advised obtaining IV access prior to interventional pain management procedures for patients with a high risk of vasovagal syncope [13,46,48]. However, no studies to our knowledge have evaluated whether IV fluid administration during interventional pain management procedures can prevent vasovagal reactions. More research should be done on the benefit of obtaining IV access prior to these procedures, both as a stand-alone preventive measure with IV fluids and in conjunction with other pharmacological agents (Table 6).

## 6. Management

Most—if not all—patients undergoing interventional pain management procedures should have vital sign monitoring, including pulse oximetry, an electrocardiogram, and blood pressure monitoring, prior to and during the procedure. This is especially important for patients who report a history of vasovagal reactions to similar procedures. For patients who develop bradycardia and/or vasodepression, the first step is to stop the procedure immediately. The patient should have a cold compress (e.g., ice pack) placed on his or her neck. The patient can then either be placed supine or in the Trendelenburg position (with the table at an angle such that the patient’s head is declined below their feet at roughly a 15–30 degree angle). The patient can also be asked to perform counterpressure techniques, such as squatting or leg crossing, which may improve venous return and cardiac output. If these conservative measures do not work, IV fluids should be started (if they were not started preoperatively) and vasoactive medications considered, such as ephedrine in 5–10 mg increments, glycopyrrolate in 0.2 mg increments, or atropine in 0.4–1.0 mg increments. If the patient continues to have SBP < 90 mmHg, MAP < 65 mmHg, or HR < 50 bpm, then he or she should be transported to an emergency department [13].

## 7. Conclusions and Future Research

Although vasovagal reactions are a rare consequence of interventional pain management procedures, they are the most common complication of such reported in the literature. They are usually benign in nature, but can have several negative effects on both the patient and provider, including aborted procedures and fear of future procedures that would otherwise help the patient. Thus, identifying the risk factors, preventing, and treating vasovagal reactions is important for outpatient pain medicine providers. The management of vasovagal reactions has been well documented in the literature, but less studied is the use of sedation, anxiolytics, and antimuscarinics for prevention. The efficacy of these preventive measures has been demonstrated in a variety of clinical settings, but more research needs to be conducted on their utility for interventional pain management procedures specifically. These preventive measures have the potential to improve patient care, especially in patients with a history of vasovagal reaction.

## Figures and Tables

**Table 1 medsci-10-00039-t001:** Search methods on PubMed for glycopyrrolate as a preventive measure for vasovagal syncope from interventional pain management procedures.

	Glycopyrrolate	Vasovagal Reaction	Interventional Spinal Procedure
Subject headings	“Glycopyrrolate”[Mesh] OR	“Syncope, Vasovagal”[Mesh] OR	“Injections, Spinal”[Mesh] OR
Textwords	Glycopyrrolate[tiab] OR Glycopyrronium[tiab] OR Cuvposa[tiab] OR Robinul[tiab]	Syncope[tiab] OR Vasovagal[tiab] OR Vagally mediated[tiab] OR Faint*[tiab] OR Dizzy[tiab] OR Dizzi*[tiab] OR Light headed[tiab] OR Hypotensive event*[tiab] OR Bradycardi*[tiab]	Spine[tiab] OR Spinal[tiab] OR Injection[tiab] OR Epidural[tiab]

**Table 2 medsci-10-00039-t002:** Incidence and risk factors of vasovagal reactions during interventional pain management procedures.

Papers	Type of Study	Procedures Studied	Relevant Findings
Kennedy et al., 2013 [8]	Retrospective analysis	8010 spinal injections from 2004 to 2009	VV rate 2.6% for all interventional pain management procedures ESI VV rate 3.5% Males twice as likely to have VV reactions as females Pts <65 yo 2.4 times higher odds of VV reaction VV rate of 3.2% if pre-procedure pain <5/10 compared to 2.2% w/those of pain >5/10
Trentman et al., 2009 [11]	Retrospective analysis	249 pts undergoing CESI or LESI from 1996 and 2005	VV rate of 8% for CESI compared to 1% for LESI
Abbasi et al., 2007 [6]	Literature review	All papers on PubMed on pts undergoing ICESI from 1996 to 2005	VV rate commonly reported at 0–4%
Botwin et al., 2003 [10]	Retrospective analysis	157 pts receiving 345 ICESI from 2000 to 2001	VV rate of 1.7% for ICESI
Schneider et al., 2014 [16]	Retrospective analysis	2642 pts undergoing 2282 TFESI from 2004 to 2009	3.5% VV reaction overall 2.7% VV reaction when performed by attending, 4.1% fellow, 5.5% resident

VV = vasovagal. ESI = epidural steroid injection. LESI = lumbar epidural steroid injection. CESI = cervical epidural steroid injection. ICESI = interlaminar cervical epidural steroid injection. TFESI = transforaminal epidural steroid injection. Pts = patients.

**Table 3 medsci-10-00039-t003:** Sedation during interventional pain management procedures.

Papers	Type of Study	Patient Demographic	Relevant Findings
Kennedy et al., 2015 [2]	Prospective cohort	3500 pts undergoing 6364 spine injections from 2004 to 2008, 134 with sedation	3.3% VV rate without sedation; 0% VV rate with sedation Pts w/prior VV history had 23.3% VV rate without sedation vs. 0% VV rate with sedation No side effects with sedation
Schaufele et al., 2011 [21]	Retrospective analysis	2494 interventional spine procedures in 2005, 1228 under conscious sedation and 1266 local anesthesia alone	No significant difference in rates of adverse events at 1 day and 3 days postop for pts undergoing local anesthesia vs conscious sedation
Diehn et al., 2013 [9]	Retrospective analysis	4432 pts undergoing 6878 TFESI, 7 with sedation, from 2006 to 2011	0.4% VV rate for all TFESI Concluded ESI safe enough without sedation
Hodges et al., 2007 [20]	Case report	2 cases of SCI from CESI done under moderate sedation	2 pts undergoing moderate sedation for CESI found to have iatrogenic spinal cord injury
Rathmell et al., 2011 [22]	ASA closed claims study	ASA malpractice closed claims from cervical pain treatments from 2005 to 2008	67% of cervical procedure claims associated w/SCI used sedation or anesthesia 25% of pts w/claims of cervical SCI under sedation were nonresponsive during procedure

ESI = epidural steroid injection. TFLESI = transforaminal lumbar epidural steroid injection. SCI = spinal cord injury.

**Table 4 medsci-10-00039-t004:** Anxiety and anxiolytic use in vasovagal reactions.

Papers	Type of Study	Patient Demographic	Relevant Findings
Ekinci et al., 2017 [15]	Prospective cohort	210 patients with planned surgery in perianal and inguinal regions	Higher scores on preoperative anxiety inventories correlated with higher likelihood of vasovagal response
Van Vlymen et al., 1999 [24]	Randomized, double-blind, placebo-controlled study	90 women undergoing needle-guided breast biopsies randomized to receive preprocedural midazolam, diazepam, or placebo	Preoperative benzodiazepines lowered anxiety levels by 55–68% No adverse effects
Ravitskiy et al., 2011 [26]	Randomized, double-blind, placebo-controlled study	44 patients undergoing Mohs surgery randomized to receive preoperative midazolam or placebo	Preoperative midazolam significantly lowered anxiety within 1 h of administration No adverse effects
Gebhardt Volker et al., 2018 [27]	Retrospective analysis	2747 patients undergoing low-dose intrathecal anesthesia from 2008 to 2017, 1291 receiving 1–2 mg preoperative midazolam	7.5% VV rate w/preoperative midazolam vs 15% VV rate with placebo No adverse effects
James et al., 2005 [28]	Retrospective analysis	19 women administered 2–4 mg lorazepam prior to stereotactic breast biopsy between 2001 and 2004, 14 of whom had prior vasovagal reaction	13 of 14 women w/prior VV reaction did not have subsequent VV reaction w/lorazepam No adverse effects

**Table 5 medsci-10-00039-t005:** Antimuscarinic use for treating and preventing vasovagal reactions.

Papers	Type of Study	Patient Demographic	Relevant Findings
Santini et al., 1999 [41]	Single-blinded, randomized, placebo-controlled trial	84 pts w/recurrent vasovagal syncope randomized to receive IV atropine (0.02 mg/kg) or placebo after “tilt test” *	Atropine significantly resolved cardioinhibitory forms of VV reaction (70% vs 22% patients), but not vasodepressor forms
Rama et al., 2012 [42]	Double-blinded, randomized, placebo-controlled trial	165 pts randomized to receive IV atropine (0.5 mg) or placebo 5 min prior to femoral arterial sheath removal	Preoperative atropine significantly lowered VV rate (2.3% vs 10.4%). VV rates not separated into cardioinhibitory, vasovagal, or mixed
Sun et al., 2017 [1]	Randomized controlled trial	25 pts w/paroxysmal A fib undergoing cryoballoon (CB) ablation randomized to receive 1 mg IV atropine or nothing before procedure	Pre-operative atropine significantly lowered vasovagal rate (4/12 vs 12/13 pts), including vasodepressor (3/12 vs 6/13 pts), cardioinhibitory (1/12 vs 3/13 pts), and mixed forms (0/12 vs 3/13 pts)
Chamchad et al., 2011 [43]	Double-blinded, randomized, placebo-controlled trial	69 women at term randomized to receive 0.4 mg IV glycopyrrolate or IV saline followed by spinal anesthesia prior to C-section	Preoperative glycopyrrolate significantly decreased episodes of bradycardia (0/34 pts vs. 6/35 pts). No difference in hypotension rates
Mirakhur et al., 1982 [40]	Randomized, placebo-controlled trial	160 children (1–14 years old) undergoing ophthalmological (squint) surgery randomized to receive atropine, glycopyrrolate, or placebo at various doses and routes of administration (IV or IM)	IV administration of glycopyrrolate or atropine significantly lowered oculocardiac reflex, a type of vasovagal syncope defined as reduction in HR by >20% Glycopyrrolate associated with smaller magnitude of tachycardia than atropine
Yang et al., 1996 [45]	Letter to editor	3 case reports of patients receiving IM glycopyrrolate prior to ophthalmological (squint) surgery	In all cases, glycopyrrolate prevented the oculocardiac reflex with no side effects of tachycardia or dry mouth

* Tilt test is a provocative maneuver for vasovagal reaction.

**Table 6 medsci-10-00039-t006:** Summary of prophylactic methods for vasovagal reactions.

Prophylactic Method	Beneficial in Prevention of Vasovagal Reactions?	Side Effects or Negative Consequences	References
Sedation	Yes	Risk of spinal cord/nerve injury Airway compromise Risk of aspiration Nausea/vomiting Allergic reactions Fatal cardiac arrhythmias Cannot operate motor vehicles shortly after procedure	Kennedy et al., 2015 [2] Diehn et al., 2013 [9] Hodges et al., 1998 [20] Schaufele et al., 2011 [21] Rathmell et al., 2011 [22]
Anxiolytics	Yes, albeit not studied for interventional pain management procedures	Cannot operate motor vehicles shortly after procedure Confusion Anterograde amnesia Agitation Increased risk of falling	Van Vlymen et al., 1999 [24] Ravitskiy et al., 2011 [26] Gebhardt et al., 2018 [27] James et al., 2005 [28]
Antimuscarinics	Yes, albeit not studied for interventional pain management procedures	Tachycardia * Dry mucous membranes Anhidrosis Urinary retention Constipation	Sun et al., 2017 [1] Mirakhur and Dundee 1980 [40] Santini et al., 1999 [41] Rama et al., 2012 [42] Chamchad et al., 2011 [43] Yang et al., 1996 [45]
IV fluids	Uncertain; not studied for interventional pain management procedures	Minimal to no side effects	Vidri et al., 2021 [13] Mahajan 2008 [46] Kamar et al., 2021 [48]

* More likely with atropine than glycopyrrolate, as reported in the literature.

## Data Availability

Not applicable.

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
