# Peer review of "Vasovagal Reactions during Interventional Pain Management Procedures—A Review of Pathophysiology, Incidence, Risk Factors, Prevention, and Management"

_medsci, 2022, doi:10.3390/medsci10030039_

Round 1

Reviewer 1 Report

The manuscript entitled " Vasovagal Reactions During Spinal Injections – A Review of  Pathophysiology, Incidence, Risk Factors, Prevention, and 3 Management" by  Malave and  Vrooman summarized the available information on vasovagal reactions in this review. In my opinion the manuscript is written very systematically by addressing all the major points and by discussing the clinical cases with their proper citations.

Author Response

Thank you for your review. We appreciate your input and time in reviewing our manuscript.

Reviewer 2 Report

In this article, the authors presented a narrative review of vasovagal reactions during spinal injections, and showed their pathophysiology, incidence, risk factors, preventions, and managements. It is an interesting and well-summarized article, however, there are some concerns which have to be clarified.

Major criticism

1.     This study focused spinal injections; however, their definitions were unclear. Most referred articles might be studies regarding epidural steroid injections. On one hand, authors cited articles about epidural anesthesia to explain the pathophysiology of vasovagal reaction during spinal injections (P3, L80). The heterogeneity of definition of spinal injections may disturb readers’ understandings.

2.     In this review, some descriptions were specific for spinal injections (a part of pathophysiology, incidence, and a part of risk factors); however, the others were not. Authors should emphasize the specific aspects for spinal injections, or rename the article title not to be misleading.

Minor criticism

1.     Vital sign monitoring (P4, L151) and IV access (P9, L320) should be described in management section rather than in prevention section, though hydration would be kept in prevention section.

Author Response

  1. Thank you for your detailed review and attention to this important detail. We agree that our definitions of spinal injections were unclear. We have changed the title and re-written the paper such that we clearly define which interventional pain management procedures we are referring to.
  2. Thank you again for drawing attention to this. As stated above, we changed our title name and were more careful in our use of “spinal injection” vs “interventional pain management procedure,” for which we provide a definition. We hope our edits make this clearer.
  3. We moved information about vital sign monitoring and IV access to the management section. Under the prevention section, we wrote a subsection on hydration and IV fluids as a possible prophylactic measure for vasovagal syncope.

Reviewer 3 Report

This review is well written and very useful for practitioners who perform the spinal injections. The contents are no problems, but there are many misprints of the reference numbers in the manuscript for example: Page 1, line 36: [2] is correct? Page 2, line 63: correct [3] to [2]. Page 3, line 78: correct [4] to [3], and so on. The authors need to correct and ascertain the reference numbers in the whole manuscript.

Minor points

Page 6, line 215: “Gebhard” may be a misprint for “Gebhardt”?

Page 8, line 292: “2012” is a misprint for “2011”.

Author Response

Thank you for your review. We corrected the mistakes you pointed out – i.e. 2011 rather than 2012 and “Gebhard” to “Gebhardt.” We also corrected our reference list and double-checked for accuracy (our prior submission had the incorrect bibliography attached, leading our reference numbers to be off).

Round 2

Reviewer 2 Report

Authors faithfully revised their work, and improved it.

I think it is suitable for publication in the present form.